# Integration of Network Pharmacology and Molecular Docking to Analyse the Mechanism of Action of Oregano Essential Oil in the Treatment of Bovine Mastitis

**DOI:** 10.3390/vetsci10050350

**Published:** 2023-05-14

**Authors:** Guangjie Cao, Jing Liu, Huan Liu, Xiaojie Chen, Na Yu, Xiubo Li, Fei Xu

**Affiliations:** 1National Feed Drug Reference Laboratories, Institute of Feed Research, Chinese Academy of Agricultural Sciences, Beijing 100081, China; 82101215576@caas.cn (G.C.);; 2Key Laboratory of Animal Antimicrobial Resistance Surveillance, Ministry of Agriculture and Rural Affairs, Beijing 100081, China

**Keywords:** oregano, bovine mastitis, network pharmacology, action mechanism

## Abstract

**Simple Summary:**

Oregano essential oil is a plant essential oil extracted from the plant oregano (Origanum vulgare), which has an antibacterial, anti-inflammatory, and antioxidant effect. Bovine mastitis is a process in which bacterial infection and inflammation co-exist. In this study, network pharmacology technology was used to explore the mechanism of oregano essential oil in the treatment of bovine mastitis, and to predict the core genes and key signaling pathways exerting therapeutic effects. The results showed that the key action targets of oregano essential oil in the treatment of dairy bovine mastitis were TNF, TLR4, ALB, IL-1β, TLR2, IL-6, IFNG, and MyD88, and the main signaling pathways were PI3K-Akt, MAPK, IL-17, NF-κ B. The molecular docking confirmed that thymol and carvacrol in oregano essential oil had a good binding ability to the core targets of TNF, IL-6, MyD88, and ALB. These results indicated that thymol and carvacrol had good anti-inflammatory effects. Therefore, oregano essential oil was considered to have a good therapeutic effect on bovine mastitis.

**Abstract:**

The active components, potential targets, and mechanisms of action of oregano essential oil in the treatment of bovine mastitis disease were investigated using network pharmacology and molecular docking approaches. The TCMSP and literature databases were examined for the main compounds in oregano essential oil. Afterward, the physical, chemical, and bioavailability characteristics of the components were evaluated. The PubChem, BATMAN, PharmMapper, and Uniprot databases were utilized to predict the target genes of the major components of oregano essential oil. Via the databases of DrugBank, OMIM, GeneCards, TTD, and DisGenet, the disease targets of bovine mastitis were discovered. We analyzed common targets and built protein–protein interaction (PPI) networks using the STRING database. Key genes were analyzed, obtained, and compound–target–pathway–disease visualization networks were created using Cytoscape. For the GO function and KEGG pathway enrichment analysis, the DAVID database was utilized. Molecular docking via Autodock Tools was utilized to evaluate the reliability of the interactions between oregano essential oil and hub targets. Thymol, carvacrol, and p-cymene are the three major components found in oregano essential oil. The potential targets (TNF, TLR4, ALB, IL-1β, TLR2, IL-6, IFNG, and MyD88) were screened according to the visual network. The enrichment analysis suggested that the major signaling pathways in network pharmacology may include PI3K-Akt, MAPK, IL-17, and NF-κ B. Molecular docking analysis shows that thymol had good docking activity with TNF, IL-6, and MyD88, carvacrol had good docking activity with TNF, and p-cymene had good docking activity with ALB. This study clarified the mechanism of action of oregano essential oil in the treatment of bovine mastitis, thus providing data supporting the potential for the use of oregano essential oil in the development of new therapeutics for bovine mastitis.

## 1. Introduction

Bovine mastitis is a worldwide problem that affects breeding. It is one of the four major diseases of bovines that seriously affects the quality and output of milk, and it causes serious economic losses [1]. Bovine mastitis is an inflammation of the breast tissue brought on by a bacterial infection, a chemical reaction, or some other type of trauma [2]. Infection with pathogenic bacteria not only causes bovine mastitis, but also leads to an inflammatory reaction [3]. Histopathological changes, changes in the physical and chemical properties of milk, and changes in the number of somatic cells are the principal symptoms of bovine mastitis [4]. In addition, bovine mastitis leads to abnormal functioning of the reproductive system, resulting in delayed estrus, decreased pregnancy rate, and increased risk of abortion [5,6]. Nowadays, antimicrobial therapy is the principal treatment for bovine mastitis. The most commonly used antibacterial drugs are antibiotics, including cephalosporins, penicillin, erythromycin, quinolones, and sulfonamides [7,8]. With the extensive use of antibiotics, the drug resistance of bovine mastitis pathogens has become increasingly serious. A study found that 83.4% (60/72) of the 72 isolated pathogenic bacteria of bovine mastitis had a multi-antibiotic resistance index higher than 0.2 [9]. Antibiotics have been reported to have little effect on the treatment of clinical or subclinical bovine mastitis caused by *E. coli*, probably due to the increasing number of cases of recurrent and persistent mastitis [10]. After antibiotic treatment, there is often a “post-bactericidal effect” in the clinic; the inflammatory reaction will last for a period of time, and the number of somatic cells in milk will still be high [11]. The post-bactericidal effect can even turn into recessive mastitis, leading to repeated attacks of bovine mastitis [12]. Essential oils are volatile compounds extracted from the flowers, leaves, rhizomes, and stems of plants; and are composed of secondary metabolites such as terpenes, phenols, alcohols, and aldehydes [13]. As a new type of plant-derived antibiotic, essential oils have the advantages of a broad antibacterial spectrum, being green, being safe, having no residue, and not being easy to produce drug resistance against [14,15]. It has been reported that the essential oil-based pharmaceutical (Phyto-Bomat), alone or in combination with antibiotics, shows promising results in the treatment of mastitis and that no breast stimulates the risk [16]. Therefore, due to the special advantages of plant essential oils, their application in the treatment of infectious diseases will become a trend.

Oregano, or Origanum (family: Lamiaceae), and Lippia (family: Verbenaceae) have many effects, including relieving exterior syndrome, clearing summer heat, and eliminating dampness [17,18]. Oregano essential oil is a pale yellow liquid extracted from oregano that has an aromatic flavor. The main components of oregano essential oil include carvacrol, thymol, isopropyl acryl toluene, rosmarinic acid, p-cymene, and caryophyllene [19,20]. Oregano essential oil has many pharmacological effects, such as antioxidant, antibacterial, anti-inflammatory, antiviral, anti-tumor, antiparasitic, and immunomodulatory effects [21,22,23]. One study found that oregano essential oil had strong antibacterial activity and anti-inflammatory effects, and that oregano essential oil inhibited and killed bacteria by inhibiting the synthesis of bacterial proteins, enterotoxin, and grape flavin [24,25]. Oregano essential oil inhibits the expression and secretion of interleukin-1β (IL-1β), interleukin-6 (IL-6), and tumor necrosis factor-α (TNF-α) of LPS-treated RAW264.7 cells, and reduces the activation of protein kinase B and nuclear factor κB in RAW264.7 cells, thereby reducing the occurrence of cellular inflammatory reactions [26]. Due to its good antibacterial and anti-inflammatory ability, it is worth exploring whether oregano essential oil has the potential to be used for the treatment of bovine mastitis.

Network pharmacology explores the relationships and mechanisms between chemical components and biological networks on a molecular level. It focuses on drug research, combining bioinformatics and network analysis, and reveals the correlations among drugs, genes, and diseases. Primary screening of action targets and signaling pathways is used to predict the mechanism of action of disease therapeutics, thereby theoretically reducing the time for and increasing the success rate of drug development [27]. In this study, network pharmacology was used to determine the mechanism of action, as well as examine oregano essential oil and its components at a molecular level, to evaluate the prospect of the development of oregano essential oil as a therapeutic for bovine mastitis.

## 2. Materials and Methods

### 2.1. Screening of Active Components Targets of Origanum Vulgare Essential Oil

The TCMSP (Traditional Chinese Medicine Systems Pharmacology) database (https://old.tcmsp-e.com/tcmsp.php (accessed on 12 August 2022)) [28] and literature review were used to search and screen [29], respectively, the main components of oregano essential oil. Potential action targets corresponding to the main components were collected from the TCMSP database. The 3D structures of the main active compounds in oregano essential oil were retrieved using the PubChem database (https://pubchem.ncbi.nlm.nih.gov/ (accessed on 12 August 2022)), and the target protein was predicted and sequenced based on the pharmacophore structure using the PharmMapper database (http://www.lilab-ecust.cn/pharmmapper/ (accessed on 12 August 2022)). Using “Bos taurus” (common bovine) as the species, the potential targets collected from the TCMSP and the PharmMapper databases were imported into the UniProt database (https://www.uniprot.org/ (accessed on 14 August 2022)) to retrieve the validated drug target genes.

### 2.2. Target Gene Screening for Disease

The target genes involved in bovine mastitis-related diseases were obtained from the GeneCard database (https://www.genecards.org/ (accessed on 16 August 2022)), DrugBank database (https://go.drugbank.com/ (accessed on 16 August 2022)), OMIM (Online Mendelian Inheritance in Man) database (https://omim.org/ (accessed on 16 August 2022)), DisGenet database (https://www.disgenet.org/ (accessed on 16 August 2022)), and TTD (Therapeutic Target Database) database (http://db.idrblab.net/ttd/ (accessed on 16 August 2022)), using Bos taurus and mastitis as screening conditions. Removing the repeated collection set of all the disease genes of the bovine mastitis obtained from each database obtained a final disease target gene.

### 2.3. Protein–Protein Interaction (PPI) Network Analysis and Construct

The drug action targets of the main components of oregano essential oil and the bovine mastitis targets were uploaded to the VENN online website (https://bioinfogp.cnb.csic.es/ (accessed on 18 August 2022)) for interactions, and the intersection target was selected. The common targets were imported into the STRING online website (https://cn.string-db.org/ (accessed on 18 August 2022)), the species was set as “Bos taurus,” and the confidence level was selected as 0.700 to establish a protein–protein interaction (PPI) network of targets for the treatment of bovine mastitis by oregano essential oil. The PPI network was analyzed using the Cytoscape 3.9.1 cytoNCA plug-in, and the core target genes were screened out and visualized.

### 2.4. GO (Gene Ontology) Function and KEGG (Kyoto Encyclopedia of Genes and Genomes) Enrichment Analysis

The core target genes were imported into the DAVID (the Database for Annotation, Visualization, and Integrated Discovery) database (https://david.ncifcrf.gov/ (accessed on 20 August 2022)), and GO function enrichment analysis and KEGG pathway enrichment analysis were performed. The GO enrichment analysis can separately describe the molecular function, cellular component, and biological process of the gene. The KEGG enrichment analysis reveals the signaling pathways involved by these genes. The GO enrichment analysis and KEGG enrichment analysis reveal which GO terms and signaling pathways with significant enrichment of intersecting targets. Cytoscape 3.9.1 was used to construct the drugs–compounds–targets–pathways–diseases network.

### 2.5. Molecular Docking Verification

The 2D structures of the active compounds of oregano essential oil were downloaded from the PubChem database, and the tertiary structures of the active compounds were optimized via Chem 3D 19.0. The ID of the core target gene was found through UniProt, and the 3D structures of the key protein were downloaded through RCSB PDB (Protein Data Bank) (https://www.rcsb.org/ (accessed on 26 August 2022)) screening. The main components of oregano essential oil and protein corresponding to the target gene were added along with water using Autodock Tools, and hydrogen was removed. Semi-flexible docking was performed via Autodock Vina, and the docking activity score between the receptor and the ligand was calculated. Binding activity was considered good if the score was less than 5.0. The structure with the lowest binding free energy was selected for visualization by PyMOL.

## 3. Results

### 3.1. Screening of Oregano Essential Oil Component Targets and Bovine Mastitis Disease Targets

According to the literature review and TCMSP database, the main components of oregano essential oil were thymol, carvacrol, and p-cymene. In total, 16 thymol, 10 carvacrol, and 1 para-cymene action targets were obtained. Target predictions for thymol, carvacrol, and p-cymene were performed using the PharmMapper database. With “NormFit” ≥ 0.3 as the screening condition, 267 potential action targets were obtained for thymol, 213 for carvacrol, and 23 for p-cymene. The potential action targets of oregano essential oil active components were standardized and converted into corresponding gene names, and 2476 target genes were obtained using the UniProt database. Using Bos taurus and mastitis as the screening conditions, 244 disease target genes for bovine mastitis were obtained from the GeneCard database, DrugBank database, OMIM database, DisGenet database, and TTD database. After duplications were removed, 217 disease targets were identified.

### 3.2. Construction of PPI Network of Key Target Proteins in the Treatment of Bovine Mastitis with Origanum Essential Oil

The target genes for the main components of oregano essential oil and the target genes for mastitis in bovines were analyzed, and 69 intersecting targets were obtained (Figure 1). The common targets were imported into the STRING database, and “Bos taurus” was selected as the species, with the confidence threshold set to “high confidence (>0.700).” The scattered nodes were hidden to construct a PPI network, including the bovine mastitis target and the main active compounds of oregano essential oil (Figure 2). The results showed that the number of nodes was 69, the number of edges was 195, the expected number of edges was 29, the average degree of nodes was 5.65, and the average local clustering coefficient was 0.553.

“Bos taurus” was selected as the species, with the confidence threshold set to “high confidence (>0.700)”. Different circles represent different targets. Different colored lines represent different relationships between genes.

The PPI network was visualized by Cytoscape 3.9.1, and the node centrality was calculated by CytoNCA. The targets are arranged according to the degree value (Figure 3). The higher the degree value, the larger the target area. The core targets with a degree ≥ 15 were selected. The results are shown in Table 1. The core targets are TNF (Tumor necrosis factor), TLR4 (Toll-Like Receptor 4), ALB (Albumin), IL-1β (Interleukin-1β), TLR2 (Toll-like receptors 2), IL-6 (Interleukin-6), IFNG (Tnterferon-gamma), and MyD88 (Myeloid Differentiation Primary Response 88).

The targets are arranged according to the degree value. Different dots represent different targets. The higher the degree value, the larger the target area. The blue dots are targets with higher degree values. Gray lines represent the connectivity between targets.

### 3.3. GO Function and KEGG Enrichment Analysis and Construction of the Component–Target–Pathway Network

The targets of the main active components of oregano essential oil for the treatment of bovine mastitis were imported into the DAVID database for GO function and KEGG pathway enrichment analysis. The results showed that there were a total of 291 GO items, including 238 BP (biological process) items, 18 CC (cell component) items, and 35 MF (molecular function) items. The main biological processes involved in the treatment of bovine mastitis with oregano essential oil include positive regulation of nitric oxide biosynthetic processes, positive regulation of MAP kinase activity, positive regulation of inflammatory response, positive regulation of RNA polymerase II promoter transcription, positive regulation of NF-κ B transcription factor activity, positive regulation of interleukin-6 production, and positive regulation of tumor necrosis factor production. Cell components mainly include the extracellular space, cell surface, cytoplasm, the external side of the plasma membrane, and the nucleus. Molecular functions mainly include protein binding, protein kinase binding, protein homodimerization activity, receptor binding, and cytokine activity (Figure 4).

Gene counts represent the number of genes contained in different BPs, CCs, and MFs. −log10 (*p* value) > 2 is a significant difference; the larger the value, the more significant the enrichment. The different GO terms are arranged according to −log10 (*p* value).

The results of the KEGG pathway enrichment analysis showed that a total of 62 pathways with significant correlation were identified, and 18 core pathways related to oregano essential oil for the treatment of bovine mastitis were screened out by retrieval (Figure 5). The oregano essential oil mainly participates in cytokine–cytokine receptor interactions, the PI3K-Akt signaling pathway, MAPK signaling pathway, IL-17 signaling pathway, NF-κ B signaling pathway, and tumor necrosis factor signaling pathway. Therefore, the main components of oregano essential oil can play a role in the treatment of bovine mastitis by regulating these signaling pathways. Drugs, their active components, action targets, related pathways, and diseases were imported into Cytoscape 3.9.1 to construct a drug–component–target–pathway–disease network (Figure 6). The figure shows the gene targets and signaling pathways acted by the main components of oregano essential oil in the treatment of bovine mastitis.

The size of the dots represents the number of genes contained in different gene pathways. A larger dot contains more genes. Different colors represent different −log10 (*p* value).

The red triangle represented oregano, the yellow diamond represented the main components of oregano, the blue diamond represented the pathway, the green square represented the target gene, and the gray line represented connectivity. This figure shows how oregano treats bovine mastitis.

### 3.4. Molecular Docking Verification

Based on the results of PPI network screening, the core targets of TNF, TLR4, ALB, IL-1β, TLR2, IL-6, IFNG, and MyD88 were selected to dock with the active components of oregano essential oil. When the combination of energy for molecular docking is less than −5.0 kcal/mol, the binding is relatively stable. The combination of energy for molecular docking is shown in Table 2. The results showed good binding between the main compounds of oregano essential oil and the core target protein (Figure 7). Among these, thymol had good docking activity with TNF, IL-6, and MyD88, carvacrol had good docking activity with TNF, and p-cymene had good docking activity with ALB. These results indicated that thymol and carvacrol had good anti-inflammatory effects in the treatment of bovine mastitis.

The figure shows the binding mode of the main components of oregano essential oil and the protein corresponding to the core targets. If the combination of the energy of molecular docking is less than −5.0 kcal/mol, the binding is relatively stable.

## 4. Discussion

In this study, three main components of oregano essential oil were selected for action target analysis, namely, thymol, carvacrol, and p-cymene. The results showed that the key targets for the pharmacological effects of the main components of oregano essential oil were TNF, TLR4, ALB, IL-1β, TLR2, IL-6, IFNG, and MyD88. Tumor necrosis factor is mainly produced by activated macrophages, NK cells, and T cells. TNF has been reported to be a pro-inflammatory factor that stimulates inflammation and plays a role in inhibiting infections and cancer processes [30]. TLR4 and TLR2 are Toll-like receptors (TLRs) that are located on the cell membrane and are indispensable parts of the innate immune response. Both TLR2 and TLR4 recruit MyD88 ligand molecules to form the TLR2/4-TIRAP-MyD88 complex, and then recruit IRAKs for phosphorylation activation, thus activating the NF-κB pathway and mediating the production of pro-inflammatory cytokines [31]. The activation of the TLR signaling pathways can induce the secretion of cytokines, including IL-6, IL-1β, and TNF-α, to participate in the inflammatory pathological response and regulate the immune response [32]. IL-6 is a 26 KD secreted protein that can stimulate the proliferation and differentiation of B cells, produce antibodies [33], and has a wide range of activities that affect the immune system, tissue stability, and metabolic processes. IL-1β and IL-6 can amplify the inflammatory response by affecting histiocytes, thus causing the accumulation of immune cells and accelerating the occurrence of inflammation [34]. The IFNG gene can encode a soluble cytokine and belongs to type II interferon. Interferon-γ is crucial for immunity against intracellular pathogens, and abnormal expression of interferon-γ is related to autoinflammation and autoimmune disease [35]. In summary, such targets as TNF, TLR4, ALB, IL-1β, TLR2, IL-6, IFNG, and MyD88 are related to the inflammatory and immune responses, indicating that the anti-inflammatory effects of oregano essential oil in bovine mastitis occur by directly acting on inflammation and by stimulating the immune response.

The GO function enrichment analysis and pathway enrichment analysis indicated that the treatment of bovine mastitis with oregano essential oil involves multiple biological processes and signaling pathways. Cytokines are small-molecule proteins synthesized and secreted by immune cells and non-immune cells. Cytokines regulate cell differentiation and immune response, as well as participate in the inflammatory response by binding to the corresponding receptors [36]. The PI3K-Akt signaling pathway may be one of the most important signaling pathways in the treatment of dairy bovine mastitis with oregano essential oil. Akt is the downstream effector of PI3K. Nrf2 is a key factor downstream of PI3K/Akt and is involved in the regulation of oxidative stress and the inflammatory response [37]. It has been found that inhibition of the PI3K/AKT/mTOR signaling pathway can reduce the secretion of IL-6, IL-8, and IL-1β, thus inhibiting the inflammatory response [38,39]. The NF-κB signaling pathway is related to immune and inflammatory reactions, as well as physiological and pathological processes in organisms. Inactivation of p53 and phosphorylation of pRb activate the NF-κB signaling pathway and cause the cells to enter the S phase, thus inducing the excessive proliferation of tumor cells [40]. Thymol in oregano essential oil has been found to inhibit the production of TNF-α, IL-6, and nitric oxide synthase in lipopolysaccharide-treated mouse mammary cells by mediating the down-regulation of AKT and NF-κB, thereby exerting an anti-inflammatory effect [41]. IL-17 is an inflammatory cytokine secreted by TH17 cells and congenital immune cells, and plays a key role in the inflammatory response and autoimmune disease [42]. A study found that IL-17 can activate the NF-κB signaling pathway, leading to the transcription of downstream signaling factors and participation in the body’s inflammatory response [43]. NO is an important signal transduction factor that is produced by L-arginine catalyzed by nitric oxide synthase (NOS) and plays an important role in the inflammatory process and immune response [44]. When oregano essential oil extract is used to treat inflammatory diseases caused by skin wounds, it significantly reduces the expression of pro-inflammatory mediators’ reactive oxygen species (ROS), inducible nitric oxide synthase (iNOS), and cyclooxygenase (COX)-2, thus reducing the inflammatory response and promoting inflammatory recovery [45]. The MAPK signaling pathway can regulate physiological and pathological processes such as the reproduction, differentiation, and inflammatory responses of cells. The p38MAPK signaling pathway is mainly involved in the inflammatory response and regulation of the response [46]. Studies have shown that the p38MAPK pathway plays an important role in the pathogenesis of bovine mastitis, and that regulation of the NF-κB and p38MAPK signaling pathways can inhibit breast inflammation [47,48]. In summary, the active components of oregano essential oil may play roles in the treatment of bovine mastitis through signaling pathways, including PI3K-Akt, MAPK, IL-17, and NF-κB or other signaling pathways involved in inflammation. Molecular docking showed that thymol had a good binding ability with the core targets TNF, IL-6, and MyD88, carvacrol had good docking activity with TNF, and p-cymene had good docking activity with ALB. These results indicate that thymol and carvacrol have good anti-inflammatory effects, and they may be considered the best choice for potential subsequent development into anti-inflammatory drugs.

## 5. Conclusions

In this study, network pharmacology and molecular docking technology were used to predict the key targets and main signaling pathways involved in the treatment of bovine mastitis with oregano essential oil. The main components of oregano essential oil were thymol, carvacrol, and p-cymene. The core targets selected by Cytoscape analysis were TNF, TLR4, ALB, IL-1β, TLR2, IL-6, IFNG, and MyD88. The GO enrichment analysis revealed that the main components of oregano oil were mainly involved in the positive regulation of MAP kinase activity, positive regulation of NF-kappaB transcription factor activity, positive regulation of tumor necrosis factor production, positive regulation of interleukin-6 production, positive regulation of RNA polymerase II promoter transcription, positive regulation of nitric oxide biosynthetic processes, and positive regulation of inflammatory response. The KEGG enrichment analysis showed that thymol, carvacrol, and paracetamol could participate in the regulation of the PI3K-Akt signaling pathway, MAPK signaling pathway, IL-17 signaling pathway, and NF-κB signaling pathway. Finally, it was proven by molecular docking that thymol has good docking activity with TNF, IL-6, and MyD88, carvacrol has good docking activity with TNF, and p-cymene has good docking activity with ALB. Therefore, thymol and carvacrol, the main components of origanum essential oil, have good anti-inflammatory abilities.

In summary, the main components of oregano essential oil can bind to proteins corresponding to targets such as TNF, TLR4, ALB, IL-1β, TLR2, IL-6, IFNG, and MyD88, and regulate PI3K-Akt signaling pathway, MAPK signaling pathway, IL-17 signaling pathway, and NF-κB signaling pathway to exert their anti-inflammatory effects. Therefore, the oregano essential oil can play a role in treating bovine mastitis. This study provides a theoretical basis for the treatment of bovine mastitis with oregano essential oil, and provides ideas for the development and utilization of other plant essential oils. It can be considered in the development of oregano essential oil or its main components as a new drug to treat bovine mastitis.

## Figures and Tables

**Figure 1 vetsci-10-00350-f001:**
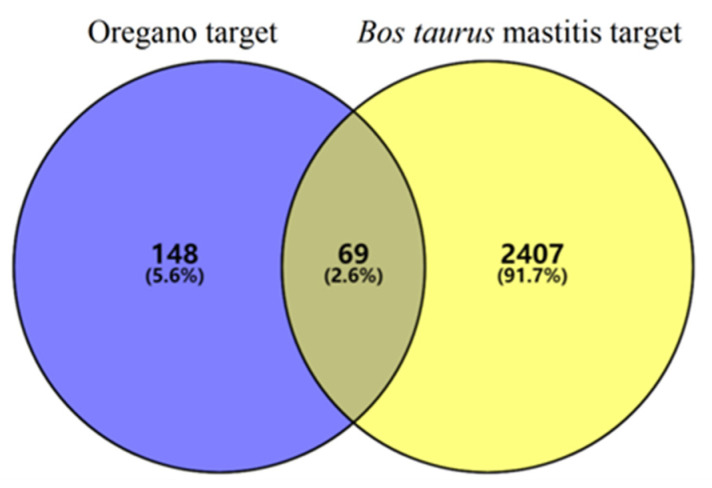
Venn diagram of the active compounds of oregano essential oil and bovine mastitis target.

**Figure 2 vetsci-10-00350-f002:**
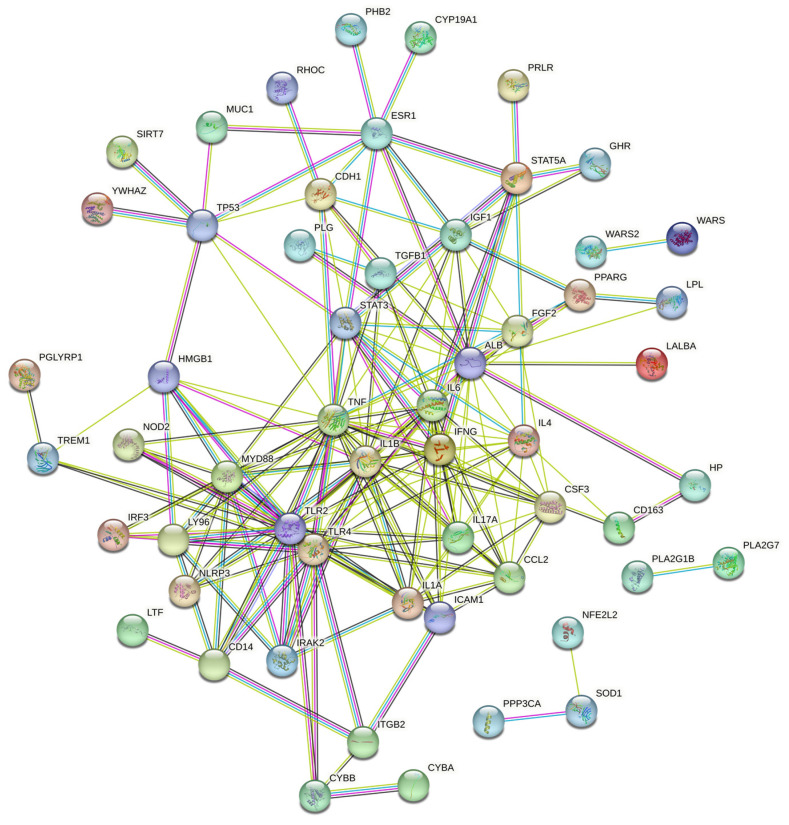
PPI network of bovine mastitis target and the main compounds of oregano essential oil.

**Figure 3 vetsci-10-00350-f003:**
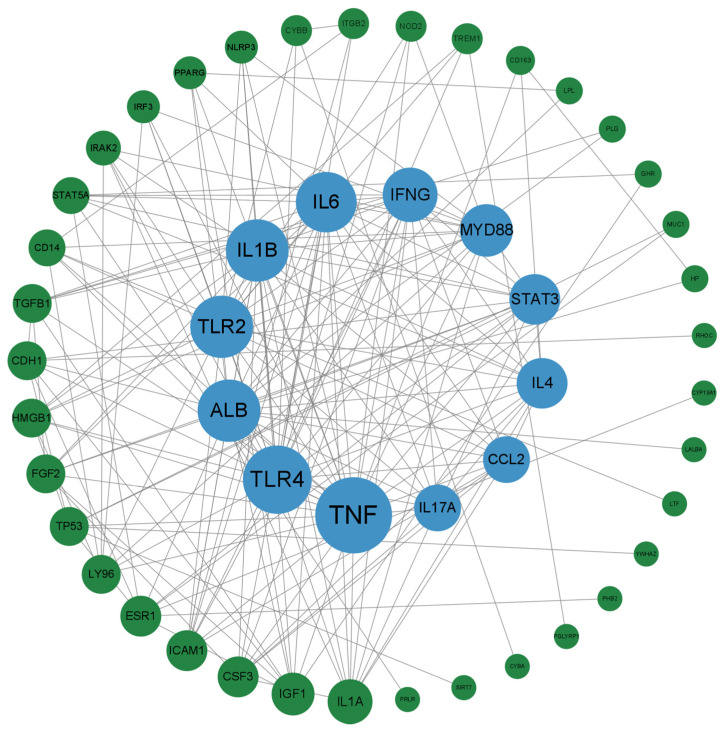
PPI network diagram based on degree value.

**Figure 4 vetsci-10-00350-f004:**
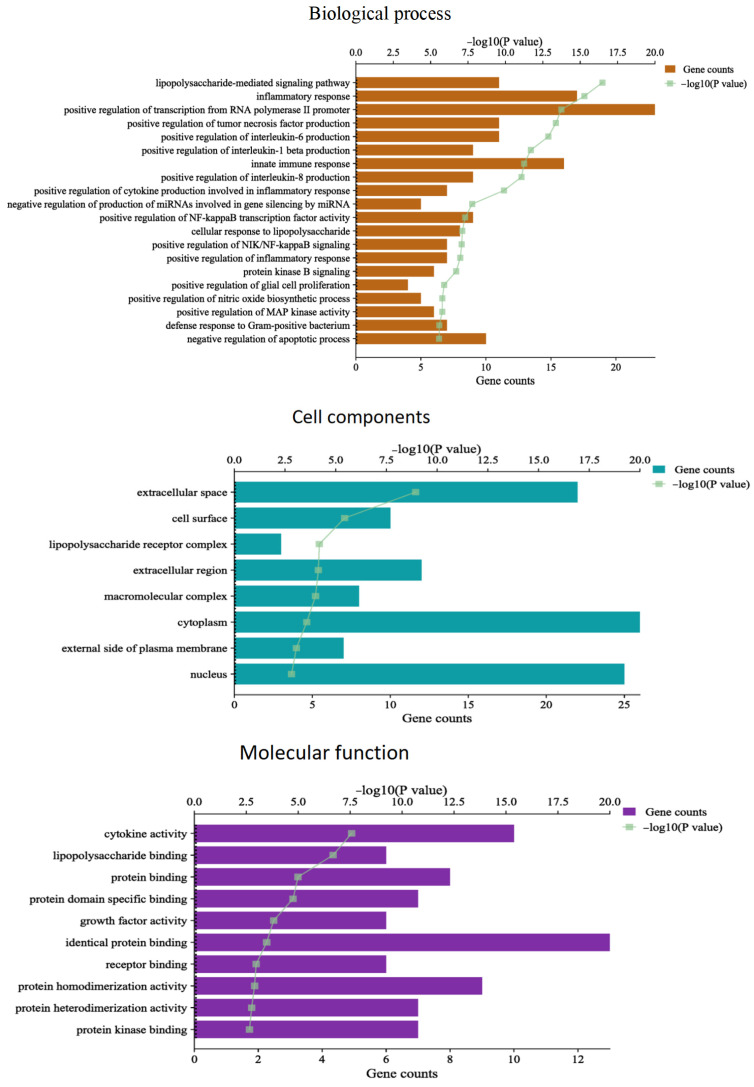
GO function enrichment analysis.

**Figure 5 vetsci-10-00350-f005:**
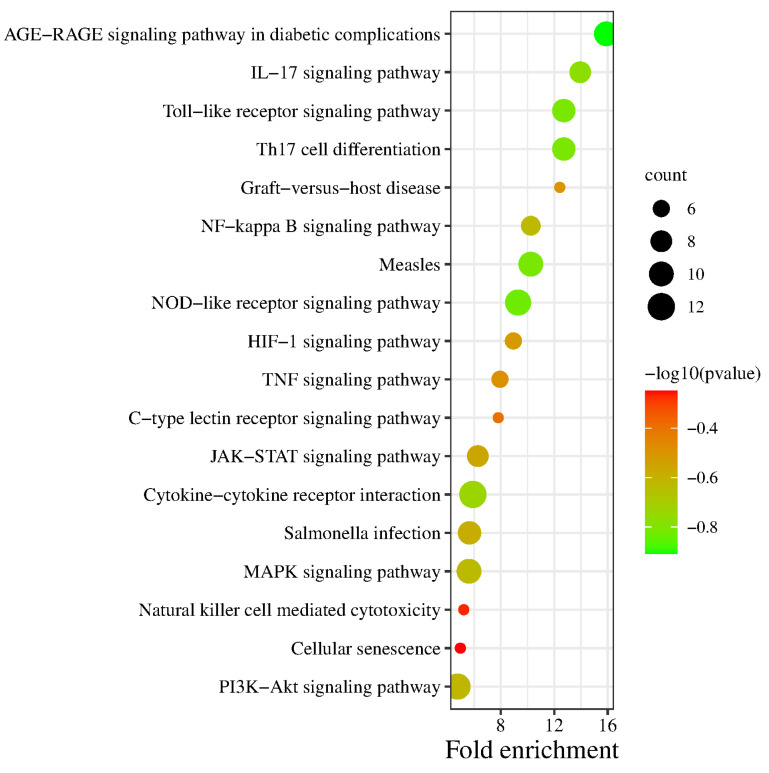
KEGG pathway enrichment analysis.

**Figure 6 vetsci-10-00350-f006:**
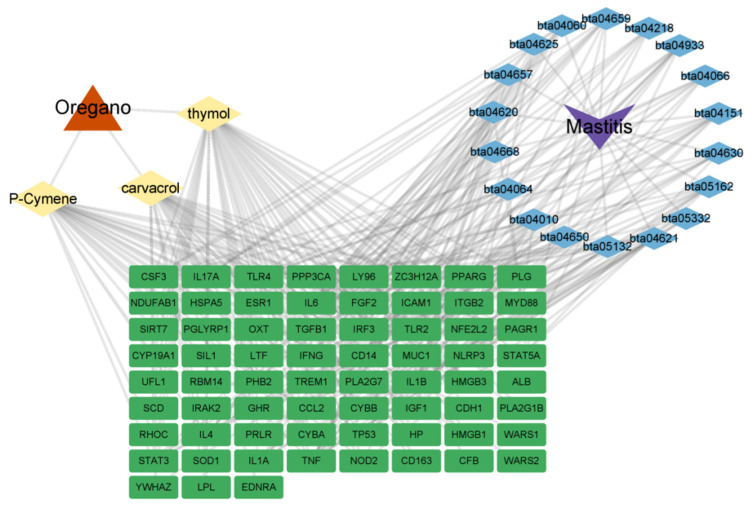
Oregano–component–target–pathway–disease network model.

**Figure 7 vetsci-10-00350-f007:**
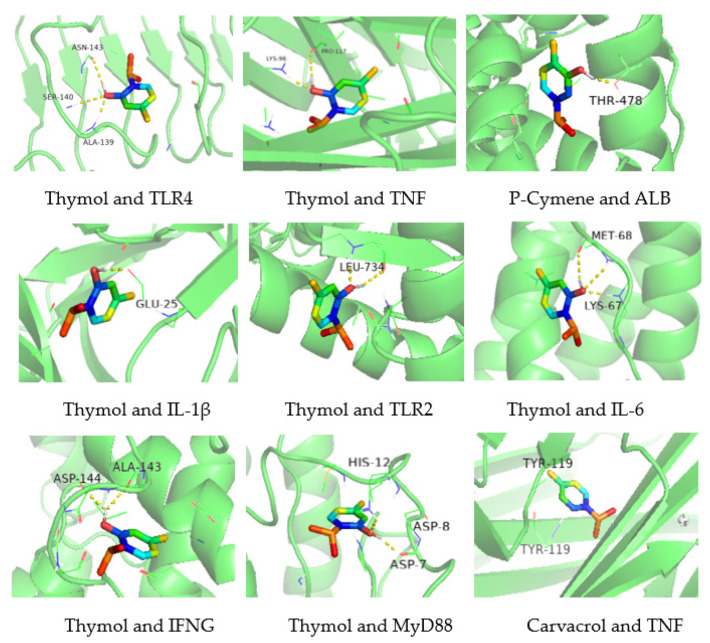
Molecular docking diagram of origanum active components and targets.

**Table 1 vetsci-10-00350-t001:** Core targets of the main components of oregano essential oil for treating bovines.

Number	Genes	Degree	UniProt ID	Names
1	TNF	27	Q06599	Tumor necrosis factor
2	TLR4	23	Q9GL65	Toll-Like Receptor 4
3	ALB	20	P02769	Albumin
4	IL-1β	20	P09428	Interleukin-1β
5	TLR2	20	Q95LA9	Toll-like receptors 2
6	IL-6	19	P26892	Interleukin-6
7	IFNG	16	P07353	Interferon-gamma
8	MyD88	15	Q599T9	Myeloid Differentiation Primary Response 88

**Table 2 vetsci-10-00350-t002:** Molecular docking score of origanum active components and targets.

Components	Targets	Combination of Energy (kcal/mol)
Thymol	TNF	−5.88
Carvacrol	TNF	−5.61
P-Cymene	ALB	−5.01
Thymol	TLR4	−4.87
Thymol	IL-1β	−4.61
Thymol	TLR2	−3.65
Thymol	IL-6	−5.11
Carvacrol	IL-6	−4.96
Thymol	IFNG	−4.51
Thymol	MyD88	−5.52

## Data Availability

Data are contained within the article.

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
