# Peer review of "Integration of Network Pharmacology and Molecular Docking to Analyse the Mechanism of Action of Oregano Essential Oil in the Treatment of Bovine Mastitis"

_vetsci, 2023, doi:10.3390/vetsci10050350_

Round 1
Reviewer 1 Report
The authors developed a relevant study for the dairy cattle production chain. It is essential to use bioinformatics techniques in order to prospect new molecules with therapeutic potential. It is important to reduce the time inside the laboratory, accelerating the production of new mastitis treatment options.
I suggest a minor revision!
Introduction Lines 44 – 57 the authors mentioned relevant aspects of mastitis. I suggest, before the next paragraph, to mention the relevance of alternative therapies, indicating the main advantages.
Methodology: Why the authors did not include a conventional antimicrobial drug in the molecular docking analysis. This is important, as a parameter for analysis.
Discussion: Line 205: paracetamol or p-cymene
Reviewer 2 Report
Dear Authors,
Your manuscript titled "Integration of Network Pharmacology and Molecular Docking to Analyse the Mechanism of Action of Oregano Oil in the Treatment of Bovine Mastitis" is addressing an interesting research topic for publication at Veterinary Sciences.
The main question addressed by the research was to use network pharmacology to determine the mechanism of action, as well as examine oregano oil and its components at a molecular level, to evaluate the prospect of the development of oregano oil as a therapeutic for bovine mastitis according to authors' aim.It is a relevant question for Veterinary Sciences due it is addressed to reduce the use of antibiotics in dairy cattle by using a new methodology to solve an important problem such as mastitis in dairies by focusing on: 1) mastitis as one of the major diseases of bovines that affects the quality and output of milk and it causes serious economic losses in the dairy sector, and 2) network pharmacology as an innovative methodology which explores the relationships and mechanisms between chemical components and biological networks on a molecular level by combining bioinformatics and network analysis to reveal the correlations among drugs, genes, and diseases to evaluate the prospect of the development in current research of oregano oil as a therapeutic strategy for mastitis due to its pharmacological effects, such as antioxidant, antibacterial, anti-inflammatory, antiviral, anti-tumor, antiparasitic, and immunomodulatory. The research conducted seems interesting due to its originality/novelity to address an important problem for dairies such as bovine mastitis. It provides new knowledge by applying network pharmacology as an original approach for exploring the underlying relationships and mechanisms between chemical components and biochemical networks to evaluate the use of essential oils (for example, oregano oil) as a therapeutic strategy for bovine mastitis. To my knowledge, no previous articles have been published to date in this topic by using network pharmacology with oregano oil for bovine mastitis. The present study predicted the key targets and main signaling pathways involved in the treatment of mastitis in dairy bovines by oregano oil using network pharmacology and molecular docking technology. It indicates that the treatment of bovine mastitis with oregano oil may be mainly achieved through such the key action targets as TNF, TLR4, ALB, IL-1β, TLR2, IL-6, IFNG 16 and MyD88, and the main signaling pathways to that were PI3K-Akt, MAPK, IL-17, NF-κB. In summary, the study provides a theoretical basis for the treatment of bovine mastitis with oregano oil which might be useful for future research to be conducted on this topic. The conclusions are consistent with the evidence and argumented presented in the text, Tables and Figures. They address in them the main question posed.
It might be acceptable for publication after minor revision. Please, see below a list of comments and suggestions to be applied by you to improve its quality before accepting it for publication.
L11 Replace 'oregano, has' by 'oregano, which has'.
L27-L28 Replace 'discov-ered' by 'discove-red'.
L29-L30 Replace 'vis-ualisation' by 'visua-lisation'.
L38-L39 Replace 'sup-porting' by 'su-pporting'.
L46 Add a blank space before `[1]'.
L47 Add a blank space before `[2]'.
L53 Add a blank space before `[3, 4]'.
L57 Add a blank space before `[5]'.
L62 Add a blank space before `[6, 7]'.
L64 Add a blank space before `[8, 9]'.
L65 Replace 'bac-teria' by 'ba-cteria'.
L66 Add a blank space before `[10, 11]'.
L68 Add a blank space before `[12]'.
L76 Add a blank space before `[13]'.
L77-L78 Replace 'exam-ine' by 'exami-ne'.
L82 Add a blank space before `[14]'.
L83 Add a blank space before `[15]'.
L103-L104 Replace 'interac-tion' by 'intera-ction'.
L120-L121 Replace 'struc-ture' by 'stru-cture'.
L131 Add a blank space after `[names]'.
L153 Replace 'higher' by 'highest' and 'larger' by 'largest'.
L157 Replace 'higher' by 'highest' and 'larger' by 'largest'.
L166-L167 Replace 'ox-ide' by 'oxi-de'.
L209-L210 Replace 'infec-tions' by 'infe-ctions'.
L210 Add a blank space before `[16]'.
L214 Add a blank space before `[17]'.
L217 Add a blank space before `[18]'.
L217-L218 Replace 'prolifer-ation' by 'prolifera-tion'.
L218 Add a blank space before `[19]'.
L222 Add a blank space before `[20]'.
L223 Replace 'expres-sion' by 'expre-ssion'.
L224 Add a blank space before `[21]'.
L234 Add a blank space before `[22]'.
L237 Add a blank space before `[23]'.
L239 Add a blank space before `[24, 25]'.
L241-L242 Replace 'ac-tivate' by 'acti-vate'.
L243 Add a blank space before `[26]'.
L246 Add a blank space before `[27]'.
L248 Add a blank space before `[28]'.
L250 Add a blank space before `[29]'.
L252 Add a blank space before `[30]'.
L256 Add a blank space before `[31]'.
L260 Add a blank space before `[32]'.
L262 Add a blank space before `[33, 34]'.
L278 Delete '6. Patents'.
L279-L280 Replace 'exper-imental' by 'experi-mental'.
L283 Replace 'article..' by 'article.'.
L284 Delete 'Please add:'.
L292-372 Review all references following MDPI's guidelines for authors.
L295 Delete '[J]'.
L297 Delete a blank space before 'effects' and delete '[J]'.
L300 Delete '[J]'.
L302 Delete '[J]'.
L303 Delete '[J]'.
L304 Replace '84-87..' by '84-87'.
L306-L307 Delete '[J]' and replace 'bioproducts proce-ssing' by 'Bioproducts Proce-ssing'.
L309 Delete '[J]'.
L311-L312 Delete '[J]' and replace 'animal sci-ence' by 'Animal Sci-ence'.
L314 Delete '[J]'.
L317 Delete '[J]' and replace 'proteomics' by 'Proteomics'.
L319 Delete '[J]'.
L323 Delete '[J]'.
L325 Delete '[J]'.
L327 Delete '[J]'.
L329 Delete '[J]'.
L330 Delete '[J]'.
L333 Delete '[J]'.
L335 Delete '[J]'.
L337 Delete '[J]'.
L339 Delete '[J]'.
L341 Delete '[J]'.
L344 Delete '[J]'.
L347 Delete '[J]'.
L350 Delete '[J]'.
L352 Delete '[J]'.
L354 Delete '[J]'.
L356 Delete '[J]'.
L359 Delete '[J]'.
L361 Delete '[J]'.
L363 Delete '[J]'.
L364 Replace 'chemical toxicology' by 'Chemical Toxicology'.
L366 Delete a blank space before 'lung' and delete '[J]'.
L369 Delete '[D]'.
L372 Delete '[D]'.
L373 Delete this line.
L374 Delete this line.
Reviewer 3 Report
This manuscripts studies the way the main active compounds present in the oregano essential oil may fight against mastitis infections in bovins.
In the introduction some references are missing.
You should include the botanical name of Oregano, and always refer to the essential oil, instead of just oil, because it's not exactly the same and could induce confussion.
The Material and methods section is well described, only you shoudl add the explanation of all the acronyms used.
The results are globally well explained, even somre figures have not been completely treated or well explained: 3, 4, 5 and 6.
The conclusion is good
More detailed corrections can be found in the attached document
